# Significance of Vascular Cell Adhesion Molecule-1 and Tumor Necrosis Factor-Alpha in HIV-Infected Patients

**DOI:** 10.3390/jcm11030514

**Published:** 2022-01-20

**Authors:** Tomasz Mikuła, Magdalena Suchacz, Mariusz Sapuła, Alicja Wiercińska-Drapało

**Affiliations:** 1Department of Infectious and Tropical Diseases and Hepatology, The Medical University of Warsaw, 02-091 Warsaw, Poland; magdalena.suchacz@wum.edu.pl (M.S.); mariusz.sapula@wum.edu.pl (M.S.); alicja.wiercinska-drapalo@wum.edu.pl (A.W.-D.); 2Warsaw’s Hospital for Infectious Diseases, 01-201 Warsaw, Poland

**Keywords:** VCAM-1, TNF-alpha, HIV, procalcitonin, CRP

## Abstract

Background. The aim of this study was the evaluation of the correlation between VCAM-1 and TNF-alpha serum concentrations and various clinical and laboratory parameters in HIV-infected patients. Methods. All included subjects were patients of the Department of Infectious and Tropical Diseases and Hepatology of the Medical University of Warsaw in Poland in the years 2014–2016. The inclusion criteria were: confirmed HIV infection, Caucasian origin, and age > 18 years old. PCT, CRP, serum HIV-1 RNA, CD4/CD8 T cell count, PCR HCV RNA, HBsAg, VCAM-1, and TNF-alpha were measured. The VCAM-1 and TNF-alpha serum levels were evaluated by ELISA. Results. Seventy-two HIV-infected patients were included (16 women and 56 men: mean age 38.7 years, 66.6% cigarette smokers, 34.7% HCV co-infected HCV, and 27.8% ART-naïve). VCAM-1 concentrations were significantly higher in HIV/HCV co-infected patients than in HIV mono-infected patients (125.6 ± 85.4 vs. 78.4 ± 58.6 ng/mL, *p* = 0.011) and ART-naïve in comparison to patients on cART (121.9 ± 76.5 vs. 69.4 ± 57.1 ng/mL, *p* = 0.003). The significant positive correlation between HCV-infection and VCAM-1 and negative correlation between cART use and VCAM-1 was confirmed in multivariate analyses. The only variable associated significantly with TNF-alpha concentration was lymphocytes T CD8+ cell count (*p* = 0.026, estimate = 0.033). Conclusions. Successful cART and HCV eradication seemed to play an important role in the reduction of endothelial dysfunction and persistent inflammation in HIV-infected patients. CD8 T cell count seemed to be one of the markers of the pro-inflammatory state in HIV-infection patients.

## 1. Introduction

The pathogenesis of the cardiovascular disease (CVD) in HIV infection is complex and not fully understood. However, it has already been confirmed that Human Immunodeficiency Virus (HIV) exacerbates atherosclerotic lesions development through a long-lasting localized inflammation linked with permanent changes in the endothelium [1].

Vascular cell adhesion molecule 1 (VCAM-1) is an adhesion molecule expressed on the activated endothelial surface, promoting the interaction of endothelial cells with leukocytes; the process being the basis for the formation of atherosclerotic plaque. It has been shown that VCAM-1 might be a biomarker of immunological diseases, cancer, and autoimmune myocarditis. Moreover, VCAM-1 is linked with the risk of heart failure development and with endothelial injury in patients with coronary artery disease and arrhythmias [2]. Finally, VCAM-1 expression can be induced by the HIV protein Tat-1 [3].

The activation of endothelial cells occurs mainly after exposure to lipids, inflammatory stimulating factors, and pro-inflammatory cytokines such as interleukin 1β (IL-1β), interleukin 6 (IL-6), and tumor necrosis factor-alpha (TNF-alpha) [4]. TNF-alpha is a cytokine synthesized by activated monocytes, macrophages, adipocytes, neutrophils, and lymphocytes. TNF-alpha binds to cells through two types of receptors, R1 and R2, causing increased insulin resistance in peripheral tissues, stimulation of phagocytosis, and increased production of CRP by the liver. Moreover, a high concentration of TNF-alpha is independently associated with an increased thickness of the carotid intima-media, which increases the risk of cardiovascular disease [5,6].

The aim of our study was to evaluate the correlation between VCAM-1 and TNF-alpha serum concentrations and various clinical and laboratory parameters in HIV-infected patients treated or untreated with combined antiretroviral therapy (cART).

## 2. Materials and Methods

All HIV-infected patients who participated in this study were under the care of the Department of Infectious and Tropical Diseases and Hepatology of the Medical University of Warsaw in Poland in the years 2014–2016. Inclusion to the study took place during one of the standard health control visits. The inclusion criteria were: confirmed HIV infection, Caucasian origin, and age > 18 years old. The exclusion criteria were: diagnosed and/or treated hypertension, cardiovascular disease, peripheral artery disease, diabetes, and use of lipid-lowering treatment. All included participants provided written informed consent. The study was approved by the Bioethics Committee of the Medical University of Warsaw and followed the principles of the Declaration of Helsinki.

Every included patient was asked about the time of HIV-infection diagnosis, the time of cART starting, the composition of cART, concomitant disorders, drug use, and cigarette smoking. We used the CDC definition of the smoker as a person who has smoked 100 cigarettes in his or her lifetime, or who currently smokes cigarettes.

Fasting blood samples were used to measure the serum level of procalcitonin (PCT) and C Reactive Protein (CRP). The serum HIV-1 viral load was measured by the standard quantitative polymerase chain reaction (PCR) by Real-Time PCR, Test Cobas 6800 HIV Roche The CD4 T cell count, CD8 T cell count, and CD4+/CD8+ ratio was determined by flow cytometry. HCV infection was assessed by the presence of serum HCV RNA, and HBV infection by the presence of serum circulating hepatitis B surface antigen. The VCAM-1 and TNF-alpha serum levels were evaluated by enzyme-linked immunosorbent assay (Human sVCAM-1/CD106 Quantikine ELISA and Human TNF-alpha Quantikine ELISA, R&D Systems, respectively). All parameters were analyzed from blood samples taken during the same control visit.

The R package (version 3.5.1) was used for all statistical calculations. Student’s *t*-test and Pearson’s correlation were used for each type of variable in the univariate analysis. In the multivariate analysis, a linear model with a stepwise elimination method guided by the *p*-value was used to determine which variables were independently associated with VCAM-1 and TNF-alpha levels. A *p*-value of <0.05 was assumed to be statistically significant. All variables used in the univariate analysis were used in the multivariate analysis.

## 3. Results

Seventy-two HIV-infected patients were included in the study: 16 women (22.2%) and 56 men (77.8%). The mean age was 38.7 years (range 21–74 years). Mean HIV infection and cART duration were 6.6 and 3.4 years, respectively. Forty-eight persons (66.6%) were cigarette smokers, 25 persons (34.7%) were co-infected with HCV (none treated against HCV before), and 11 persons (15.3%) were co-infected with HBV. The baseline characteristics of the studied group are shown in Table 1.

Twenty studied patients (27.8%) were ART-naïve, and 52 patients (72.2%) were on effective cART. Every ARV-treated patient had undetectable serum HIV viral load (<50 copies/mL) at the time of study and was treated among guidelines obligatory in 2014–2016. All ARV-treated patients obtained nucleoside reverse transcriptase inhibitors (NRTIs) as a therapy backbone, plus different third agent protease inhibitors (PIs) were used in 41 patients (lopinavir boosted with ritonavir-18, atazanavir boosted with ritonavir-16, darunavir boosted with ritonavir-7), non-nucleoside reverse transcriptase inhibitors (NNRTIs) in 10 patients (efavirenz -7, nevirapine -3), integrase strand transfer inhibitors (INSTIs) in 3 patients (raltegravir-2, dolutegravir-1), and CCR5 inhibitor in 1 patient. All HBV/HIV co-infected patients obtained tenofovir/emtricitabine combination as NRTIs’ backbone.

The results of univariate and multivariate associations of VCAM-1 concentrations with different clinical, immunological, and biochemical variables are shown in Table 2, Table 3 and Table 4. Estimates of the influence of various parameters on VCAM-1 concentrations are shown along with *p*-values for significant variables. All variables used in the univariate analysis were used in the multivariate analysis, but only variables that were associated with VCAM-1 concentrations in a statistically significant manner in the last step of the stepwise analysis are shown.

The results of univariate associations of TNF-alpha concentrations with different clinical, immunological, and biochemical variables are shown in Table 5 and Table 6. Estimates of the influence of various parameters on TNF-alpha concentrations are shown along with *p*-values for significant variables. All variables used in the univariate analysis were used in the multivariate analysis. The only variable that was associated with TNF-alpha concentration in a statistically significant manner in the last step of the stepwise analysis was lymphocytes T CD8+ cell count (*p* = 0.026, estimate = 0.033). Other variables had no important correlation with serum TNF-alpha concentration.

## 4. Discussion

The aim of our study was to evaluate the correlation between VCAM-1 and TNF-alpha serum concentrations and various clinical and laboratory parameters in HIV-infected patients treated or untreated with combined antiretroviral therapy (cART). We observed that VCAM-1 concentrations were significantly higher in HIV/HCV co-infected patients and ART-naïves, as the only variable associated significantly with TNF-alpha concentration was lymphocytes T CD8+ cell count.

HIV-infected patients had higher values of VCAM-1 than HIV-uninfected patients [7]. The HIV-linked serum VCAM-1 concentration may be a marker of HIV-dependent endothelial activation and HIV infection progression [8]. Effective cART caused control of HIV replication, and as a result may have influenced VCAM-1 levels. Francisci et al., showed that the VCAM-1 concentration decreased significantly 24 months after cART initiation, although it was still above the norm [9]. In a study by Maggi et al., 119 HIV-infected ART-naïve patients with CD4+ T cell counts less than 200 cells/µL were examined. The observed VCAM-1 serum concentration was consistently raised during the 12th month after cART initiation [10]. In our group of patients, the average duration of antiretroviral therapy exceeded 24 months, and we also showed a significant decrease in mean serum VCAM-1 concentrations in patients with effective cART in comparison to ART-naïve patients. This observation may indirectly confirm the protective role of effective cART on cardiovascular risk in the HIV-infected population.

Moreover, in our study we showed significantly higher serum VCAM-1 concentrations in HIV/HCV co-infected patients in comparison to HIV mono-infected subjects. This observation is coherent with several previous studies [11,12]. HCV may enhance chronic inflammation, and consequently HIV/HCV co-infection is linked with higher levels of systemic inflammation markers, monocyte activations, and endothelial dysfunction [11,12]. This correlation between HIV, HCV, and pro-inflammatory processes has very important clinical relevance, because it may be linked with the degree of liver fibrosis in HIV/HCV co-infected patients [13]. Recently, it has been shown that HCV eradication after effective treatment with direct acting antivirals (DAAs) or after liver transplantation diminished significantly, and even often normalized serum markers’ levels of endothelial activation [14,15]. However, in HIV/HCV co-infected patients the situation is more complex, and even successful HCV eradication does not always lead to the normalization of systemic inflammation and endothelial dysfunction conditions, especially in persons with already-increased liver fibrosis. This phenomenon is probably linked with persistent HIV infection and its pro-inflammatory influence, independent of the HCV infection state [16].

In our study, we did not show the differences and correlations between serum TNF-alpha level and ART status or HCV co-infection in HIV-infected patients. However, we showed a significant positive correlation between TNF-alpha concentration and CD8 T cell count. CD8 T cells are key mediators of antiviral and antitumor immunity. CD8 T cells—by secretion of different cytokines as interleukin-2 (IL-2), interferon-gamma (IFN-γ), and TNF-alpha as their direct cytotoxicity—exhibit anti-HIV activity. HIV chronic persistence plays a major role in chronic activation of HIV-specific cytotoxic CD8 T-cells and production of these pro-inflammatory cytokines [17]. Sinha et al., showed that the CD8 T cell count and TNF-alpha concentration were associated with microvascular dysfunction in HIV-infected patients [18]. Similar results were obtained by Grome et al., who showed that circulating CD8 T cell activation may impair arterial smooth muscle relaxation, leading to arterial vascular disease in HIV-infected persons on cART [19].

Moreover, recently Behrens et al., presented data showing that HIV-specific CD8 TNF-alpha expression decreased with increasing CD4 T-cell counts, suggesting reduced activation of HIV-specific CD8 T-cells due to effective cART. This is the next important proof that virologically and immunologically successful cART may diminish pro-inflammatory state and cardiovascular risk in HIV-infected patients. Unfortunately, this positive influence diminishes progressively with age [20]. In our analysis, TNF-alpha was not associated with cART use. This was maybe caused by too small of a studied group and a relatively high number of ART-naïve patients (20; 27.8%).

Our study has some limitations. Firstly, this study is a single-center study with a relatively small number of participants. Secondly, in our analyses, we did not take into account the meantime from cART initiation as time from HIV RNA undetectability. Finally, we could not perform an ultrasonographic assessment of the carotid intima-media thickness that would allow for a better assessment of changes in carotid intima-media, their correlation with the obtained results, and finally a more accurate analysis of cardiovascular risk among our patients.

## 5. Conclusions

We demonstrated that HCV co-infection and deficiency of effective cART are linked with higher serum concentrations of VCAM-1 being a biomarker of endothelial dysfunction. Moreover, we observed a positive correlation between CD8 T cell count and pro-inflammatory cytokine TNF-alpha serum level.

Successful cART and HCV eradication seemed to play an important role in the reduction of endothelial dysfunction, persistent inflammation, and cardiovascular risk in HIV-infected patients. CD8 T cell count seemed to be one of the markers of the pro-inflammatory state in HIV-infection patients.

## Figures and Tables

**Table 1 jcm-11-00514-t001:** Baseline patients’ characteristics with means and standard deviation (SD).

	Mean	SD
CRP (mg/L)	21.63	44.07
PCT (ng/mL)	0.61	3.58
VCAM-1 (ng/mL)	1824.00	1370.40
TNF-alpha (pg/mL)	64.55	64.99
Lymphocytes T CD4+ (cell/µL)	318.06	229.73
Lymphocytes T CD4+ (%)	28.74	15.76
Lymphocytes T CD8+ (cell/µL)	786.96	566.49
Lymphocytes T CD8+ (%)	70.29	14.71
Lymphocytes T CD4+/CD8+ ratio	0.48	0.40

CRP-C—reactive protein; PCT—procalcitonin; VCAM-1—vascular cell adhesion molecule-1; TNF-alpha—tumor necrosis factor alpha.

**Table 2 jcm-11-00514-t002:** Pearson’s correlations with VCAM-1 concentrations–univariate analysis.

Variable	R	R^2^	*p*
Age (years)	−0.04	0.002	0.264
Lymphocytes T CD4+ (cell/µL)	−0.17	0.030	0.164
Lymphocytes T CD8+ (cell/µL)	−0.12	0.015	0.328
Lymphocytes T CD4+/CD8+ ratio	−0.03	0.001	0.807

**Table 3 jcm-11-00514-t003:** Mean VCAM-1 concentrations (ng/mL) for various categorical variables.

	Yes		No		
Variable	Mean	SD	Mean	SD	*p*
Males	95.9	74.2	86.1	62.4	0.646
HCV	125.6	85.4	78.4	58.6	0.011
cART	69.4	57.1	121.9	76.5	0.003
Smoking cigarettes	102.5	75.3	76.2	60.7	0.160
CRP > 5 mg/L	120.4	75.8	72.1	60.2	0.006
PCT > 0.05 ng/mL	155.8	50.4	78.1	67.5	<0.001

HCV—hepatitis C virus; cART—combined antiretroviral therapy; CRP-C—reactive protein; PCT—procalcitonin.

**Table 4 jcm-11-00514-t004:** VCAM-1 concentrations—a multivariate analysis.

Variable	*p*	Estimate
HCV-positive	0.036	35.2
cART	0.009	−41.5
PCT > 0.05 ng/mL	0.006	56.3
Other variables	>0.05	Not significant

HCV—hepatitis C virus; cART—combined antiretroviral therapy; PCT—procalcitonin.

**Table 5 jcm-11-00514-t005:** Pearson’s correlations with TNF-alpha concentrations—univariate analysis.

Variable	R	R^2^	*p*
Age	−0.03	0.001	0.941
Lymphocytes T CD4+ (cell/µL)	−0.03	0.001	0.764
Lymphocytes T CD8+ (cell/µL)	0.28	0.076	0.026
Lymphocytes T CD4+/CD8+ ratio	−0.15	0.021	0.248

**Table 6 jcm-11-00514-t006:** Mean TNF-alpha concentrations (pg/mL) for various categorical variables.

	Yes		No		
Variable	Mean	SD	Mean	SD	*p*
Males	65.3	74.1	72.0	43.2	0.740
HCV	55.9	33.5	72.1	79.1	0.370
cART	63.3	35.7	71.1	93.1	0.647
Smoking cigarettes	64.8	71.5	70.8	61.7	0.738
CRP > 5 mg/L	60.3	34.3	72.2	86.3	0.487
PCT > 0.05 ng/mL	44.4	17.8	72.5	74.6	0.185

HCV—hepatitis C virus; cART—combined antiretroviral therapy; CRP-C—reactive protein; PCT—procalcitonin.

## Data Availability

The data presented in this study are available on request from the corresponding author.

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
