# Peer review of "Significance of Vascular Cell Adhesion Molecule-1 and Tumor Necrosis Factor-Alpha in HIV-Infected Patients"

_jcm, 2022, doi:10.3390/jcm11030514_

Round 1

Reviewer 1 Report

The authors investigate sVCAM-1 and TNF-a in HIV positive patients. Both biomarkers are related to clinical and immunological parameters. The pathogenesis of CVD in HIV infection is not fully understood and analysis of biomarkers might add new insights. However, the topic is not novel as it has been shown many times that immune activation (including CRP, sVCAM etc) is increased in PLWH and does not return to normal levels despite initiation of ART. The methods are not clear (see my comments below), and I also do not see how the findings contribute to the field (either research or clinical decision making). Although, there are some interesting findings on the relationship with HCV and CD8+ cells. I would recommend that the authors define the use of their findings and provide clear directions how these findings can be explored further to understand the pathophysiology of atherosclerosis in HIV, or in risk prediction.

Abstract:

Conclusion does not follow the results. How it this: ‘Our findings may help in new treatment strategies focused on inflammation and cytotoxic mechanisms of CD8 lymphocytes affecting immune system in infected patients based on the results? Especially of CD8 cells it’s only reported that TNF-alfa levels correlate with CD-8. The conclusion refers to a plural effect ‘mechanisms’. I also don’t see / understand what is meant practically with how the findings can be translated in new treatment strategies, and how determination of VCAM and TNFa may support the panel of test. It’s quite vague the way it is formulated currently.

Introduction

PLHIV is not introduced.

Line 59: ‘to monitor .. lipid disorders’. That’s not the point of the discussion and I suggest to take it out.

Line 61: what is ‘this phenomenon’? No biomarker thus far has been proven to reflect atherosclerosis reliably, at least not beyond the weight of traditional CVD risk factors.

Methods

Line 65: immunological markers. In the abstract these are called: ‘biomarkers’. Please choose one naming.

The methods are not reproducible. What were inclusion criteria? What was the time period of the study? Were all consecutive patients enrolled unless they met an exclusion criteria? What was the research site (what is ‘our department’)? Why were patients admitted? (was there an intercurrent infection for example?)

Why was active smoking defined as at least 5 packyears? What is someone smoked 10 cigarettes a day?

Where was the sample size based on?

The analysis is also no reproducible: what variables were entered in the regression analysis? What variables in the multivariate analysis?

Please indicate if informed consent was obtained from the participants.

Discussion

Lines 123-127. What were the findings of this study?

Line 137: please check if thickness of inner carotid artery membrane is correct. The normal measurement measures carotid intima-media thickness, so the thickness of the inner 2 layers of the artery.

Lines 144-145: what is ‘an association between carotid intima media thickness inflammatory markers’?

Conclusion

Line 172-173: ‘In HIV/HCV coinfected patients VCAM-1 serum concentration can indicate potential 172 endothelial dysfunction’. How does this follow from your results? HCV in HIV infected patients is associated with increased levels of VCAM-1. VCAM-1 can indicate endothelials dysfunction in all patients. Please rephrase. And also be more inclusive. ART use is also associated with VCAM-1 levels.

Line 173-174: ‘Determination to of the patients. Please remove or rephrase. Its very vague now. How can it complement the treatment? What does it indicate? To check for HCV (in case of sVCAM-1) for example? If it’s meant to improve CVD health, how would it help in risk stratification?

Line 175 – 177 our results etc: how? What new treatment strategies? Anti-inflammatory treatment has been tested in PLWH aiming to reduce CVD risk, with no beneficial risk-benefit balance. Please see for example:

Ridker PM, et al; CANTOS Trial Group. Antiinflammatory therapy with canakinumab for atherosclerotic disease. N Engl J Med. 2017;377:1119–1131. doi: 10.1056/NEJMoa1707914

Ridker PM, et al; CIRT Investigators. Low-dose methotrexate for the prevention

of atherosclerotic events. N Engl J Med. 2019;380:752–762. doi:

10.1056/NEJMoa1809798

General

A thorough rewriting of the article is needed. All staccato sentences need to be adjusted. For example, line 86: ‘cigarette smoking 48/72 (66.6%)’ should be something like: cigarette smoking was xx.

Author Response

Open Review

Comments and Suggestions for Authors

The authors investigate sVCAM-1 and TNF-a in HIV positive patients. Both biomarkers are related to clinical and immunological parameters. The pathogenesis of CVD in HIV infection is not fully understood and analysis of biomarkers might add new insights. However, the topic is not novel as it has been shown many times that immune activation (including CRP, sVCAM etc) is increased in PLWH and does not return to normal levels despite initiation of ART. The methods are not clear (see my comments below), and I also do not see how the findings contribute to the field (either research or clinical decision making). Although, there are some interesting findings on the relationship with HCV and CD8+ cells. I would recommend that the authors define the use of their findings and provide clear directions how these findings can be explored further to understand the pathophysiology of atherosclerosis in HIV, or in risk prediction.

 Abstract:

Conclusion does not follow the results. How it this: ‘Our findings may help in new treatment strategies focused on inflammation and cytotoxic mechanisms of CD8 lymphocytes affecting immune system in infected patients based on the results? Especially of CD8 cells it’s only reported that TNF-alfa levels correlate with CD-8. The conclusion refers to a plural effect ‘mechanisms’. I also don’t see / understand what is meant practically with how the findings can be translated in new treatment strategies, and how determination of VCAM and TNFa may support the panel of test. It’s quite vague the way it is formulated currently. – all abstract was changed and thoroughly rewritten

 The manuscript has been thoroughly rewritten.

Introduction

PLHIV is not introduced – introduction was rewritten and PLHIV was removed from manuscript

Line 59: ‘to monitor .. lipid disorders’. That’s not the point of the discussion and I suggest to take it out. - introduction was rewritten and this fragment concerning lipid disorders was removed from manuscript

Line 61: what is ‘this phenomenon’? No biomarker thus far has been proven to reflect atherosclerosis reliably, at least not beyond the weight of traditional CVD risk factors. - introduction was rewritten and this fragment was removed from manuscript

Methods

Line 65: immunological markers. In the abstract these are called: ‘biomarkers’. Please choose one naming. – abstract and methods section were thoroughly rewritten and one naming was introduced

The methods are not reproducible. What were inclusion criteria? What was the time period of the study? Were all consecutive patients enrolled unless they met an exclusion criteria? What was the research site (what is ‘our department’)? Why were patients admitted? (was there an intercurrent infection for example?) - methods section were thoroughly rewritten, inclusion criteria, time period of the study, exclusion criteria, the research site and reason of admission were introduced to methods section

Why was active smoking defined as at least 5 packyears? What is someone smoked 10 cigarettes a day? - We used CDC definition of smoker as a person who has smoked 100 cigarettes in his or her lifetime or who currently smokes cigarettes. The definition in manuscript was rewritten for being more comprehensible.

Where was the sample size based on? - As no standard of clinically significant change in VCAM-1 and TNF-alpha exists, we decided to base our sample size considerations on comparisons with similar, previously published studies.

The analysis is also no reproducible: what variables were entered in the regression analysis? What variables in the multivariate analysis?- This is a very valid and astute criticism – thank you for pointing this out. We have modified the Methods section and descriptions below some of the tables.

Please indicate if informed consent was obtained from the participants. – informed consent was obtain from every participant, the information about consent was introduced to Methods section

Discussion – all Discussion section was thoroughly rewritten and different studies and references were introduced

Lines 123-127. What were the findings of this study? – section was rewritten

Line 137: please check if thickness of inner carotid artery membrane is correct. The normal measurement measures carotid intima-media thickness, so the thickness of the inner 2 layers of the artery. – section was rewritten

Lines 144-145: what is ‘an association between carotid intima media thickness inflammatory markers’? – section was rewritten

Conclusion - all Conclusion section was thoroughly rewritten

Line 172-173: ‘In HIV/HCV coinfected patients VCAM-1 serum concentration can indicate potential 172 endothelial dysfunction’. How does this follow from your results? HCV in HIV infected patients is associated with increased levels of VCAM-1. VCAM-1 can indicate endothelials dysfunction in all patients. Please rephrase. And also be more inclusive. ART use is also associated with VCAM-1 levels. - section was rewritten

Line 173-174: ‘Determination to of the patients. Please remove or rephrase. Its very vague now. How can it complement the treatment? What does it indicate? To check for HCV (in case of sVCAM-1) for example? If it’s meant to improve CVD health, how would it help in risk stratification? - section was rewritten

Line 175 – 177 our results etc: how? What new treatment strategies? Anti-inflammatory treatment has been tested in PLWH aiming to reduce CVD risk, with no beneficial risk-benefit balance. Please see for example: - section was rewritten

Ridker PM, et al; CANTOS Trial Group. Antiinflammatory therapy with canakinumab for atherosclerotic disease. N Engl J Med. 2017;377:1119–1131. doi: 10.1056/NEJMoa1707914

Ridker PM, et al; CIRT Investigators. Low-dose methotrexate for the prevention

of atherosclerotic events. N Engl J Med. 2019;380:752–762. doi:

10.1056/NEJMoa1809798

General

A thorough rewriting of the article is needed. All staccato sentences need to be adjusted. For example, line 86: ‘cigarette smoking 48/72 (66.6%)’ should be something like: cigarette smoking was xx. - all article was thoroughly rewritten and all staccato sentences were adjusted

Submission Date

08 November 2021

Date of this review

30 Nov 2021 21:44:13

Reviewer 2 Report

Mikula et al. aimed to find novel biomarkers to identify PLWH at higher risk for subclinical endothelial disfunction. Thus, they investigated correlations between serum concentrations of immune activation (TNF-α) and endothelial activation (VCAM-1) markers on one side, and anamnestic and laboratory parameters on the other side. Statistical analysis included Student’s t test, Pearson correlation and stepwise multivariate analysis. The Authors conclude that their findings endorse the use of TNF-α and VCAM-1 in clinical practice, specifically in the contest of HIV/HCV co-infection; and support pharmacological research on therapeutical approach targeting inflammation and CD8+-related cytotoxicity.

The study could be an useful contribution to pre-clinical research on HIV infection related inflammation among PLWH, but I several concerns listed below. Also, the paper requires extensive editing of English language and style.

Introduction

Major concerns:

·         At line 36-37 Authors say “The active Tat-1 protein within the endothelial cells of HIV-infected patients can be considered an early marker of atherosclerosis”. I suggest to modify the sentence because  it could be misleading for the readers, for the following reasons:

within: Tat-1 is a HIV soluble protein which can be secreted by HIV infected cells, circulate within blood and be able to enter uninfected cells where can exert the role of trans-activating transcriptional factor. Using the chunk within the endothelial cells can lead the readers to think that the endothelial cells are directly infected by HIV, that is not the case.

early marker: Tat-1 secretion by HIV infected cells happen regardless the presence or absence of endothelial damage, thus it shouldn’t be considered a biomarker of atherosclerosis.

  • At line 62 Authors say “we are not able to assess this phenomenon”. This sentence come out after an overview about factors influencing cardiovascular risk among PLWH, and it is not clear to which phenomenon the Authors refer to, I suggest to clarify.
  • Authors are encouraged to split the Introduction section in paragraph and specifically address topics that are of high relevance for the further Results and Discussion section. For example, the Authors dedicate broader space to role of Tat-1 protein rather than to VCAM-1, which is one of the main topic of the manuscript; I suggest to give mayor relevance to VCAM-1.

Minor concerns:

  • At line 18, Authors report the median (not mean) age, even if they state they used descriptive statistics for normally distributed data, please verify it.

At the same line, it is not clear what does “(range-74 years).” mean.

  • At line 39 the acronym for High Performance Anion-Exchange Chromatography (HPAEC) is reported, but it is never used later in the manuscript and should be removed.
  • At line 50 the word “thickness” is redundant and should be removed.

Methods

Major concerns:

  • Authors describe the statistical tests used, which are parametric tests used for normally distributed variables. Nonetheless, sample size is low, and I am unsure that all the listed variables have normal distribution. Was normality assessed with specific tests (e.g., Shapiro-Wilk test)?

If not, Authors should assess for normality and then use non-parametric tests (Wilcoxon test, Spearman correlation) for non-normally distributed variables.

If yes, Authors should declare it in the manuscript and could provide the results of the used test, at least during the current review phase.

  • The inclusion criteria are not described.

The exclusion criteria should be commented to explain the reason why they were chosen among others.

Among the exclusion criteria appears cardiovascular diseases. Was peripheral artery disease (PAD) included? And why?

  • It seems to me that Authors chose exclusion criteria in order to avoid confounding factor which can influence atherosclerosis pathogenesis. If so, it is not clear why cigarette smoke was not included among the exclusion criteria. Authors should comment on that. What is known in the literature about the cigarette smoke influence on VCAM-1 endothelial expression? Was this factor taken into account?

Also, how was the “active smoker” definition chosen? Was it arbitrary or supported by scientific literature?

  • Authors should describe the assay the used to perform the laboratory tests cited (at least for HIV-RNA, VCAM-1 and TNF-α), in order to allow other researchers to replicate their results.

Minor concerns:

·         The R package was used for all statistical calculations. Authors should declare the version of the R package used.

Results

Major concerns:

·         Authors declare the presence of 20 ART-naïve PLWH in their sample. Then, in the tables they show the categorical variable “Effective cART”. It is not clear whether “Effective cART-YES” refers to all PLWH on cART (meaning that the “Effective cART-NO” is composed solely by the 20 ART-naïve PLWH), or whether “Effective cART-YES” refers to PLWH on cART and viral load below the detectable range (which is not specified) (meaning that the “Effective cART-NO” is composed by the 20 ART-naïve PLWH + PLWH on cART and detectable viral load).

This is important, because it informs on the effect of cART alone and/or unctrolled HIV infection despite ongoing cART on the serum concentration of the outcome variables.

Also, Authors give high emphasis to the comparison between effective ART PLWH and ART-naïve PLWH in the Discussion section, that makes this point even more important.

.         At line 85-86 Authors say “Mean HIV infection and treatment duration were 6.6 and 3.4 years respectively”. What is reason for such discrepancy between time from HIV diagnosis and time from cART initiation? Did the Authors try to correlate the time from HIV diagnosis to cART initiation (that is the time between diagnosis and cART start) to serum concentrations of the outcome variables?

·         Authors provide mean time from cART initiation but not time from HIV-RNA undetectability. This variable should be taken into account.

Minor concerns:

  • Authors are not showing results about HIV viral load (VL), which is listed among the collected variable in the methods section.
  • What about the HIV/HBV-coinfected people? Was their HBV infection monitored and treated at the time of the study?
  • Table 1 shows different acronyms compared to the main text (e.g., AspAT, AlAT). Acronyms should be consistent throughout the manuscript.
  • Why Table 2 shows correlations for a few variables and not all the variables that the Athors state they collected in the Method section?
  • The multivariate analysis applied to only one variable (Table 7) is useless and could be removed.

Discussion

Major concerns:

  • There is a major confusion between the comments on the current study results and the comments on other Authors’ results (e.g., lines 118-121). Authors are encouraged to split the section into paragraphs and assess the topic one by one, clearly identifying what results/comments belong to the current study and what are belonging to other studies.

Also, Authors should focus more on the interpretation of their results first, stating whether they confirm or not their hypothesis, and then compare their results with other Authors’ results.

  • Limitations of the study should be inserted in this section.
  • At lines 123-127 Authors describe the study design of Maggi et al [10] but do not provide the study results nor comment on them.
  • At line 132-133 the Authors compare their results (coming from a cross-sectional study) to the results of a longitudinal study of Calz et al. that is not an optimal comparison.
  • At lines 148-149 Authors state that their results confirm the results of another study which verified associations between CRP, VCAM-1 and carotid intima media thickness; although, the Authors did not perform ultrasonographic assessment of the carotid intima-media thickness.

  • The sentence at line 152-153 needs extensive revision because it is not clear what the Authors want to point out.

Conclusion

The conclusions of the study are at odds with the limitations:

·         In HIV/HCV coinfected patients VCAM-1 serum concentration can indicate potential endothelial dysfunction.
Authors did not assess for endothelial disfunction nor for cardiovascular risk; so that, it is hard to state.

·         Our results can have an influence of the new treatment strategies to reduce inflammation and cytotoxic mechanisms of CD8+ lymphocytes affecting immune system in HIV- infected patients.

Correlation is not causation. Such assumption (the usefulness of pharmacologically targeting CD8-mediated-citotoxicity) is not supported by appropriate results in the current manuscript.

Author Response

Open Review

Comments and Suggestions for Authors

Mikula et al. aimed to find novel biomarkers to identify PLWH at higher risk for subclinical endothelial disfunction. Thus, they investigated correlations between serum concentrations of immune activation (TNF-α) and endothelial activation (VCAM-1) markers on one side, and anamnestic and laboratory parameters on the other side. Statistical analysis included Student’s t test, Pearson correlation and stepwise multivariate analysis. The Authors conclude that their findings endorse the use of TNF-α and VCAM-1 in clinical practice, specifically in the contest of HIV/HCV co-infection; and support pharmacological research on therapeutical approach targeting inflammation and CD8+-related cytotoxicity.

The study could be an useful contribution to pre-clinical research on HIV infection related inflammation among PLWH, but I several concerns listed below. Also, the paper requires extensive editing of English language and style.

Introduction

Major concerns: - all Introduction section was thoroughly rewritten

At line 36-37 Authors say “The active Tat-1 protein within the endothelial cells of HIV-infected patients can be considered an early marker of atherosclerosis”. I suggest to modify the sentence because  it could be misleading for the readers, for the following reasons: - this sentence was removed from rewritten manuscript

within: Tat-1 is a HIV soluble protein which can be secreted by HIV infected cells, circulate within blood and be able to enter uninfected cells where can exert the role of trans-activating transcriptional factor. Using the chunk within the endothelial cells can lead the readers to think that the endothelial cells are directly infected by HIV, that is not the case. - this sentence was removed from rewritten manuscript

early marker: Tat-1 secretion by HIV infected cells happen regardless the presence or absence of endothelial damage, thus it shouldn’t be considered a biomarker of atherosclerosis. - this sentence was removed from rewritten manuscript

  • At line 62 Authors say “we are not able to assess this phenomenon”. This sentence come out after an overview about factors influencing cardiovascular risk among PLWH, and it is not clear to which phenomenon the Authors refer to, I suggest to clarify. – all section was rewritten
  • Authors are encouraged to split the Introduction section in paragraph and specifically address topics that are of high relevance for the further Results and Discussion section. For example, the Authors dedicate broader space to role of Tat-1 protein rather than to VCAM-1, which is one of the main topic of the manuscript; I suggest to give mayor relevance to VCAM-1. – Introduction section was rewritten and splitted in paragraphs. Paragraphs concerning VCAM-1 and TNF-alpha were introduced, fragments concerning Tat-1 protein, lipid disorders and ARV treatment were removed from manuscript

Minor concerns:

  • At line 18, Authors report the median (not mean) age, even if they state they used descriptive statistics for normally distributed data, please verify it. - This was a typing error in the text that was copied into the abstract – we meant to present the mean age. Thank you for pointing this out and we are very sorry for this error. We have verified all the other calculations and luckily found no other errors.

At the same line, it is not clear what does “(range-74 years).” mean. – range of age of studied population was specified and introduced to Results section

  • At line 39 the acronym for High Performance Anion-Exchange Chromatography (HPAEC) is reported, but it is never used later in the manuscript and should be removed. – this acronym was removed from manuscript
  • At line 50 the word “thickness” is redundant and should be removed. - the word “thickness” was  removed

Methods

Major concerns:

  • Authors describe the statistical tests used, which are parametric tests used for normally distributed variables. Nonetheless, sample size is low, and I am unsure that all the listed variables have normal distribution. Was normality assessed with specific tests (e.g., Shapiro-Wilk test)?

If not, Authors should assess for normality and then use non-parametric tests (Wilcoxon test, Spearman correlation) for non-normally distributed variables.

If yes, Authors should declare it in the manuscript and could provide the results of the used test, at least during the current review phase. - We disagree with this approach. The central limit theorem states, that the distribution of the mean approximates the normal distribution with high enough sample sizes, which permits the use of parametric techniques on such samples. What exactly “high enough sample size” means in practice may be a contentious issue, but most authors suggest using a cut-off of about 30 observations per group – a condition that was satisfied in our study. Our strategy to preferentially use parametric tests and specifically to use parametric tests in this case is encouraged by statisticians specialized in biomedical research: Ghasemi and Zahediasl[1], le Cessi et al.[2], Van Buren and Herring [3] or Mishra et al.[4] – see references.

  1. Ghasemi, A.; Zahediasl, S. Normality Tests for Statistical Analysis: A Guide for Non-Statisticians. Int J Endocrinol Metab 2012, 10, 486–489, doi:10.5812/ijem.3505.
  2. le Cessie, S.; Goeman, J.J.; Dekkers, O.M. Who Is Afraid of Non-Normal Data? Choosing between Parametric and Non-Parametric Tests. European Journal of Endocrinology 2020, 182, E1–E3, doi:10.1530/EJE-19-0922.
  3. Van Buren, E.; Herring, A.H. To Be Parametric or Non‐parametric, That Is the Question: Parametric and Non‐parametric Statistical Tests. BJOG: Int J Obstet Gy 2020, 127, 549–550, doi:10.1111/1471-0528.15545.
  4. Mishra, P.; Pandey, C.; Singh, U.; Keshri, A.; Sabaretnam, M. Selection of Appropriate Statistical Methods for Data Analysis. Ann Card Anaesth 2019, 22, 297, doi:10.4103/aca.ACA_248_18.

  • The inclusion criteria are not described. – the inclusion criteria was introduced to Methods section

The exclusion criteria should be commented to explain the reason why they were chosen among others. – The exclusion criteria was introduced to Methods section, they were chosen among others as intern part of cardiovascular disease and metabolic syndrome

Among the exclusion criteria appears cardiovascular diseases. Was peripheral artery disease (PAD) included? And why? – peripheral artery disease was included in exclusion criteria as one of the type of peripheral cardiovascular disease

  • It seems to me that Authors chose exclusion criteria in order to avoid confounding factor which can influence atherosclerosis pathogenesis. If so, it is not clear why cigarette smoke was not included among the exclusion criteria. Authors should comment on that. What is known in the literature about the cigarette smoke influence on VCAM-1 endothelial expression? Was this factor taken into account? – yes, we chose exclusion criteria in order to avoid confounding factor which  can influence atherosclerosis pathogenesis. It has been already shown that smoking is linked with higher cardiovascular risk. However, smoking cigarettes is widespread among HIV-infected patients and we had problems with inclusion of appropriate number of subjects to our study. Endothelial dysfunction precedes atherosclerosis and smoking is a well-known risk factor for the development of endothelial dysfunction. Though, data are divergent. For example in recent study, Delgado et al. in smokers, higher concentrations of sICAM-1, sE-selectin sP-selectin, but lower concentrations of sL-selectin and sVCAM-1, were detected compared to never-smokers. A direct association with mortality was found for levels of sICAM-1, sVCAM-1 and vWF regardless of smoking [Delgado GE, Krämer BK, Siekmeier R, Yazdani B, März W, Leipe J, Kleber ME. Influence of smoking and smoking cessation on biomarkers of endothelial function and their association with mortality. Atherosclerosis. 2020 Jan;292:52-59. doi: 10.1016/j.atherosclerosis.2019.11.017. Epub 2019 Nov 15. PMID: 31783198]. As a result, we would like to assess the correlation of TNF-alpha and VCAM-1 with smoking status in our HIV-infected patients. Consequently, we showed no differences in TNF-alpha and VCAM-1 concentrations in smoking in comparison to non-smoking HIV-infected patients as no correlation between studied variables.  

Also, how was the “active smoker” definition chosen? Was it arbitrary or supported by scientific literature? - We used CDC definition of smoker as a person who has smoked 100 cigarettes in his or her lifetime or who currently smokes cigarettes. The definition in manuscript was rewritten for being more comprehensible.

  • Authors should describe the assay the used to perform the laboratory tests cited (at least for HIV-RNA, VCAM-1 and TNF-α), in order to allow other researchers to replicate their results. – all assays used to perform VCAM-1, TNF-alpha serum concentrations and serum HIV viral load were described in the Methods section

Minor concerns:

  • The R package was used for all statistical calculations. Authors should declare the version of the R package used. - We have added the version of the R package that was used to the manuscript (3.5.1).

Results

Major concerns:·         Authors declare the presence of 20 ART-naïve PLWH in their sample. Then, in the tables they show the categorical variable “Effective cART”. It is not clear whether “Effective cART-YES” refers to all PLWH on cART (meaning that the “Effective cART-NO” is composed solely by the 20 ART-naïve PLWH), or whether “Effective cART-YES” refers to PLWH on cART and viral load below the detectable range (which is not specified) (meaning that the “Effective cART-NO” is composed by the 20 ART-naïve PLWH + PLWH on cART and detectable viral load). – in our study effective cART refers to all PLWH on cART because every ARV-treated patient involved to our study had HIV viral load below the detectable range (< 50 copies/ml) – in the tables the categorical variable ‘effective cART’ were changed for ,cART’ and suitable information concerning HIV viral load and effectiveness of cART in our studied population was introduced to Methods section

This is important, because it informs on the effect of cART alone and/or unctrolled HIV infection despite ongoing cART on the serum concentration of the outcome variables. – as it has been already mentioned all included patients had effective cART, suitable information was introduced to Methods section

Also, Authors give high emphasis to the comparison between effective ART PLWH and ART-naïve PLWH in the Discussion section, that makes this point even more important. – all Discussion section was rewritten

At line 85-86 Authors say “Mean HIV infection and treatment duration were 6.6 and 3.4 years respectively”. What is reason for such discrepancy between time from HIV diagnosis and time from cART initiation? Did the Authors try to correlate the time from HIV diagnosis to cART initiation (that is the time between diagnosis and cART start) to serum concentrations of the outcome variables? - This discrepancy is actually commonplace in observational cohorts, as guidelines and clinical practice have changed with time considerably in terms of cART initiation. Roughly pre-2015 not all patients received cART upon diagnosis, as risk of cART toxicity and no known clearly proven benefit of early cART for patients shifted the risk-benefit ratio towards delayed cART. In that era initiation was guided mainly by immune status, with mostly patients with a clearly demonstrable immune deficit (e.g. opportunistic disease, low CD4+ cell count etc.) given cART. As new generations of cART medication were progressively developed, with better tolerability (less gastrointestinal adverse reactions, less myelosuppression, less influence of serum cholesterol, etc.) and new evidence of a clear benefit of early (that is in the few weeks after diagnosis) cART initiation in a few landmark trials (e.g. the START Study), clinical guidelines and practice have incorporated cART initiation for all people living with HIV, regardless of immune status (roughly post-2015 – at least in resource-rich settings such as Europe or North America).

In other words, patients in our study that were diagnosed pre-2015 were offered cART only once evidence of immune deficit was found (which could take years after diagnosis), and patients diagnosed post-2015 were offered cART immediately after diagnosis. Some (a minority in our opinion) of patients diagnosed post-2015 did not consent to cART for some time after the diagnosis.

We did not correlate time from HIV diagnosis to cART initiation with serum concentrations of the outcome variables, for two main reasons. Firstly, due to the above outlined reasons patients with ‘delayed cART’ (cART initiation guided by immune deficiency or cART delayed due to patients wish) and patients with ‘immediate cART’ (cART as soon as medically viable, usually within 1 month of diagnosis) have a lot of known and unknown characteristics in which they differ, which are also likely confounding factors: longer time since diagnosis, more exposure to pre-2015 cART medication, more exposure to intravenous drug use in the past, older age etc. This could potentially be alleviated by multivariate analysis, but with the number of already known relevant confounding factors this would require a patient sample many times the one we had at our disposal.

Secondly, in the aforementioned START Trial immediate cART initiation (as opposed to deferred cART initiation) was associated with lower serum inflammatory marker concentrations (such as VCAM, IL-6 or D-dimers). This would make any inferences from the correlation the time from HIV diagnosis to cART initiation with serum concentrations of the outcome variables very problematic: if there was a correlation, then it would be of no scientific value, as this is already assumed based on the results of the START Trial; if there is no correlation or the correlation is inverse, then, due to our knowledge of the results of the START Trial, the presence of some unknown variable or a mix of variables confounding the results would have to be assumed.

Baker, J.V.; Sharma, S.; Grund, B.; Rupert, A.; Metcalf, J.A.; Schechter, M.; Munderi, P.; Aho, I.; Emery, S.; Babiker, A.; et al. Systemic Inflammation, Coagulation, and Clinical Risk in the START Trial. Open Forum Infectious Diseases 2017, 4, ofx262, doi:10.1093/ofid/of

Authors provide mean time from cART initiation but not time from HIV-RNA undetectability. This variable should be taken into account. - We have not collected data on this variable, as the vast majority of treated people living with HIV reach undetectability within 1-6 months, which is a minor amount of time compared to the presumed duration of HIV infection in these patients. This variable is usually not reported in similar studies. We included this information in Limitations section.

Minor concerns:

  • Authors are not showing results about HIV viral load (VL), which is listed among the collected variable in the methods section. – all included patients had undetectable HIV viral load and were on effective cART, appropriate information was added to Methods section 
  • What about the HIV/HBV-coinfected people? Was their HBV infection monitored and treated at the time of the study? – all HIV/HBV co-infected patients included to our study was treated with TDF/FTC as a part of their cART
  • Table 1 shows different acronyms compared to the main text (e.g., AspAT, AlAT). Acronyms should be consistent throughout the manuscript. – Acronyms were changed in Table 1 and are consistent throughout the manuscript.

Why Table 2 shows correlations for a few variables and not all the variables that the Authors state they collected in the Method section? - We decided to include variables that are relevant to the characterisation on the sample in table 2, but only perform inferential statistics on variables we deemed relevant. The rationale for this was that we wanted to decrease the likelihood of a type I error due to an excessive number of inferential tests. The results of all the statistical analyses that were done were included in the tables and in the text. Moreover, we have rewritten all manuscript and removed unnecessary variables from the text.

The multivariate analysis applied to only one variable (Table 7) is useless and could be removed. - The multivariate analysis per se was done on all the variables included in the univariate analysis according to the stepwise elimination method guided by the p value. In this method, in each “step” an analysis is performed on a selection of variables (in our case: all the variables included in the univariate analysis) and the variable with the highest p value is eliminated from further analysis. This is done iteratively until every remaining variable has a p value of less than the significance threshold (in our case: 0.05) and these variables are shown as the results. In the case of the analysis shown in table 7, only lymphocyte CD8+ count was shown to be siginificantly associated with TNF-alpha concentrations independently from other variables, which was pointed out in the “other variables” row. However, we have removed Table 7 from the manuscript and the multivariate analysis results were presented in a single phrase in Results section.  

Discussion

Major concerns:

  • There is a major confusion between the comments on the current study results and the comments on other Authors’ results (e.g., lines 118-121). Authors are encouraged to split the section into paragraphs and assess the topic one by one, clearly identifying what results/comments belong to the current study and what are belonging to other studies. - All Discussion section has been rewritten and splitted into paragraphs by topic one by one clearly identifying our results in comparison to other new studies.

Also, Authors should focus more on the interpretation of their results first, stating whether they confirm or not their hypothesis, and then compare their results with other Authors’ results. - All Discussion section has been rewritten. We focused more on the interpretation of our results in comparison to new other studies.

  • Limitations of the study should be inserted in this section. – Limitations section was added to Discussion section.
  • At lines 123-127 Authors describe the study design of Maggi et al [10] but do not provide the study results nor comment on them. – Study results were commented in new version of Discussion section.
  • At line 132-133 the Authors compare their results (coming from a cross-sectional study) to the results of a longitudinal study of Calz et al. that is not an optimal comparison. – Calz’s study was removed from the Discussion section after revision of manuscript
  • At lines 148-149 Authors state that their results confirm the results of another study which verified associations between CRP, VCAM-1 and carotid intima media thickness; although, the Authors did not perform ultrasonographic assessment of the carotid intima-media thickness. – this study was removed from new version of Discussion section. The fact that we could not perform ultrasonographic assessment of the carotid intima-media thickness in our studied patients was included in Limitations section.
  • The sentence at line 152-153 needs extensive revision because it is not clear what the Authors want to point out. – all Discussion section was rewritten

Conclusion – all Conclusion section was rewritten and presently is totally different from original version of manuscript

The conclusions of the study are at odds with the limitations: - all Conclusion section was rewritten and presently is totally different from original version of manuscript

  • In HIV/HCV coinfected patients VCAM-1 serum concentration can indicate potential endothelial dysfunction.
    Authors did not assess for endothelial disfunction nor for cardiovascular risk; so that, it is hard to state. - all Conclusion section was rewritten and presently is totally different from original version of manuscript
  • Our results can have an influence of the new treatment strategies to reduce inflammation and cytotoxic mechanisms of CD8+ lymphocytes affecting immune system in HIV- infected patients.

Correlation is not causation. Such assumption (the usefulness of pharmacologically targeting CD8-mediated-citotoxicity) is not supported by appropriate results in the current manuscript. - all Conclusion section was rewritten and presently is totally different from original version of manuscript

Submission Date

08 November 2021

Date of this review

25 Nov 2021 22:40:03

Round 2

Reviewer 1 Report

The authors are commended for improving the quality and clarity of the manuscript substantially. I do have some remaining concerns. 

Abstract:

Much better, but at current there is no methods. The figures mentioned in the methods are results (description of the study population). Methods should include the setting, inclusion criteria and measurements taken.

Introduction

Line 103. What is ‘the inner layer of the internal carotid intima media’? The carotid artery has, like all arteries, three layers: adventitia, media and intima. Do you refer to an increase in intima-media thickness? If so, please rephrase.

Lines 116-117: ‘which increases the risk of blood flow disorders and thromboembolic complications’. What is a ‘blood flow disorder’? If this is about an increased intima media thickness, the main concern is that an increased intima-media thickness is directly related to the risk of overt CVD. Please rephrase.

Methods

Much clearer now. What’s not clear yet is who was approached to participate. Where this all patients attending care in the HIV clinic between 2014-2016? Or random? If so, based on what? Availability of researchers, specific recruitment days or something like this? Please clarify.

Page 8, line 158: ‘All blood samples from every patient were taken at the same time’. Does this mean that all participants came to the clinic at the same day to have their blood taken? Or did recruitment take place on a single day? Or was it the same time of the day for all participants?

Discussion

A discussion should start with a summary of your main findings, corresponding to your research question/aim. This is missing currently.

Lines 348-350: Nice literature overview. However, in your analysis TNF-a was not associated with ART use. Please reflect how that is possible given what is found in literature.

Lines 355-356: ‘Finally, we had no possibility to perform ultrasonographic assessment of the carotid intima-media thickness.’ Please explain how this would have strengthened your findings.

Conclusion

Line 364. Typo, it should be: ‘seem to play an important role’

The authors provide a clear an meaningful understanding of the importance of the relationship between HVC, ART and sVCAM. However, what is the importance of the finding that CD8+ cell count is related to TNF-a?

Author Response

Comments and Suggestions for Authors

The authors are commended for improving the quality and clarity of the manuscript substantially. I do have some remaining concerns. 

Abstract:

Much better, but at current there is no methods. The figures mentioned in the methods are results (description of the study population). Methods should include the setting, inclusion criteria and measurements taken. -  This section was rewritten

Introduction

Line 103. What is ‘the inner layer of the internal carotid intima media’? The carotid artery has, like all arteries, three layers: adventitia, media and intima. Do you refer to an increase in intima-media thickness? If so, please rephrase. This sentence was  rewritten.

Lines 116-117: ‘which increases the risk of blood flow disorders and thromboembolic complications’. What is a ‘blood flow disorder’?- It was removed from the manuscript.

 If this is about an increased intima media thickness, the main concern is that an increased intima-media thickness is directly related to the risk of overt CVD. Please rephrase. This sentence was rewritten.

Methods

Much clearer now. What’s not clear yet is who was approached to participate. Where this all patients attending care in the HIV clinic between 2014-2016? Or random? If so, based on what? Availability of researchers, specific recruitment days or something like this? Please clarify. We added more specific pieces of information about our patients.

Page 8, line 158: ‘All blood samples from every patient were taken at the same time. Does this mean that all participants came to the clinic at the same day to have their blood taken? Or did recruitment take place on a single day? Or was it the same time of the day for all participants? We added more specific pieces of information about our patients.

Discussion

A discussion should start with a summary of your main findings, corresponding to your research question/aim. This is missing currently. We added our main findings at the start.

Lines 348-350: Nice literature overview. However, in your analysis TNF-a was not associated with ART use. Please reflect how that is possible given what is found in the literature. We suggested the probable reasons why in our analysis TNF-a was not associated with ART use.

Lines 355-356: ‘Finally, we had no possibility to perform ultrasonographic assessment of the carotid intima-media thickness.’ Please explain how this would have strengthened your findings. We described wider the influence of lack of USG assessment of the carotid intima-media thickness on our results.

Conclusion

Line 364. Typo, it should be: ‘seem to play an important role’ This sentence was rewritten.

The authors provide a clear an meaningful understanding of the importance of the relationship between HVC, ART and sVCAM. However, what is the importance of the finding that CD8+ cell count is related to TNF-a? This sentence was rewritten.

With our many, many kind regards

Tomasz Mikuła MD, PhD.

Reviewer 2 Report

The manuscript revised by Author is accettable for the publication, in my opinion

Author Response

Dear Reviewer

Thanks so much for your acceptance of our manuscript.

With very kind regards Tomasz Mikuła MD, PhD.